# Realising the Environmental Potential of Vertical Farming Systems through Advances in Plant Photobiology

**DOI:** 10.3390/biology11060922

**Published:** 2022-06-16

**Authors:** Matthieu de Carbonnel, John M. Stormonth-Darling, Weiqi Liu, Dmytro Kuziak, Matthew Alan Jones

**Affiliations:** 1Oxfarm Developments, 4125 Riehen, Switzerland; matthieu@oxfarm.com (M.d.C.); dmytro.kuziak.emba-s18@said.oxford.edu (D.K.); 2UK Urban AgriTech (UKUAT) Ltd., Liverpool L1 0AF, UK; admin@ukuat.org; 3School of Molecular Biosciences, University of Glasgow, Glasgow G12 8QQ, UK; liuweiqi811@gmail.com

**Keywords:** circadian, controlled environment agriculture, demand side management, chronobiology

## Abstract

**Simple Summary:**

Vertical farming systems (VFS) have great potential for improving crop productivity but are energy-intensive, since light, temperature, and humidity each need to be controlled. In this review, we consider the challenges of incorporating renewable energy into VFS and highlight how light spectra, intensity, and daylength can be varied to influence the quality of crops. We propose that insights from plant photobiology can be utilised to optimise energy efficiency in this rapidly evolving sector.

**Abstract:**

Intensive agriculture is essential to feed increasing populations, yet requires large amounts of pesticide, fertiliser, and water to maintain productivity. One solution to mitigate these issues is the adoption of Vertical Farming Systems (VFS). The self-contained operation of these facilities offers the potential to recycle agricultural inputs, as well as sheltering crops from the effects of climate change. Recent technological advancements in light-emitting diode (LED) lighting technology have enabled VFS to become a commercial reality, although high electrical consumption continues to tarnish the environmental credentials of the industry. In this review, we examine how the inherent use of electricity by VFS can be leveraged to deliver commercial and environmental benefits. We propose that an understanding of plant photobiology can be used to vary VFS energy consumption in coordination with electrical availability from the grid, facilitating demand-side management of energy supplies and promoting crop yield.

## 1. Introduction

Human activities are contributing to global warming. In 2019, humans emitted 54 billion tonnes of carbon dioxide equivalent (CO_2_eq), of which 17 billion tonnes (31%) came from agrifood systems [1]. In developed nations, most of these emissions originate from pre- and post-production (farm gate) processes (e.g., inputs, processing, transport, retail, and waste), whereas in major developing nations, emissions tend to arise from the adoption of new farmland [1]. The agrifood system will therefore play a critical role in the climate change challenge and is increasingly recognised as such by consumers [2]. In addition to these environmental considerations, humanity also needs to provide sufficient food for an increasing global population (9.7 billion by 2050 [3]). Global crop demand is consequently expected to increase by 25–70% in this timeframe despite the limited availability of additional arable land [4,5].

As we address the demand for additional food, it will be important to consider the environmental cost of the agricultural industry. As an example, for developed nations, the UK’s agrifood value chain (including agriculture and downstream delivery chains) contributes 35% of the country’s territorial greenhouse gas (GHG) emissions (158 MtCO_2_eq, [6]). The UK government has developed a framework to reduce agrifood emissions and reach its legally binding agreement of net zero by 2050 [7]. Within this framework, controlled environment agriculture (CEA) represents an important area of funded innovation and has the potential to drive both climate change mitigation and economic and social value creation via the transformation of local agrifood systems [8].

CEA can range from low degrees of environmental control (e.g., polytunnels) to sophisticated, fully isolated facilities. These are sometimes defined separately as sunlit and sunless farms with different terminologies. This review focuses on sunless CEA operations, which we will define as vertical farming systems (VFS) [9]. We have chosen VFS because they are the most energy-consuming form of CEA and hold high potential for improving relative energy consumption, carbon emissions, and financial costs. Although energy-intensive, VFS offer alternatives to mitigate several environmental problems currently facing conventional agricultural practices. The controlled environment in VFS facilitates precision agriculture, eliminating crop protection chemicals and reducing precious inputs such as water and fertilisers [9]. Indeed, cultivation technologies using the multi-layer stacking enabled by VFS (e.g., the nutrient film technique) during lettuce growth can reduce water use by 96% compared with the traditional, non-recirculating methods often deployed in a glasshouse environment. While recirculating methods can be used in glasshouses, the absence of vertical stacking still results in an unfavourable ratio between water use and production per unit of area in VFS compared with glasshouse growth [10]. VFS also offers reduced utilisation of outdoor spaces for farmland and limits post-farm-gate emissions when situated close to urban areas [9,11]. The integration of VFS in urban supply chains is often cited as a way to transform and optimise supply chains, and holds significant benefit potential such as reductions in waste and transportation costs (time, energy), as well as increasing the quality of the consumed end product [12].

Companies working in this sector are building the financial, environmental, social and corporate governance frameworks necessary for the development of this new market [13]. Despite their potential environmental and socio-economic benefits, the current adoption of VFS is limited by the high capital expenditure and substantial energy expenditure associated with the infrastructure construction and operation of such high-control crop production systems [14]. In this review, we highlight how plant photobiology knowledge can be applied in VFS to utilise the increasing production and consumption of intermittent renewable energy, whilst also optimising crop yield and quality.

### 1.1. Defining Sources of Energy Usage and Carbon Emissions in VFS

A powerful way to measure the environmental impacts of crop production systems is life cycle assessment (LCA) [15]. Multiple types of environmental impact exist, such as water overuse, chemical pollution, and climate change GHGs. We focus here on electrical energy usage, which is typically studied and managed under a “carbon accounting” framework [16]. As illustrated in Figure 1, LCAs take not only activities at the farm into account, but also pre-farm (“upstream”) and post-farm (“downstream”) activities [17].

The carbon impact of the different activities is modelled and summed across the life cycle of a product (for instance, kg equivalent of CO_2_) and normalised to an output of the system (for example, kg of sellable lettuce). For a vertical farm, we might combine these to produce a universal metric such as CO_2_eq per kg of sellable produce. In this way, LCAs allow for a comparison of the impacts of complex systems with the same functions and outputs, such as indoor farms that use different input materials, operations, produce, and sale and delivery methods. This approach can also be used to optimise different characteristics such as spatial efficiency or growth speed, all within the context of seeking to minimise emissions. Energy use within VFS varies greatly depending on the crop grown and the system utilised. For products grown in VFS, the industry’s 2021 census found a range from less than 1 kWh to 1000 kWh, and an average of 38.8 kWh per kg of product [13]. When using the UK government’s tool to calculate emissions from UK electricity [7], this corresponds to 8.3 kg of CO_2_eq per kg of product. Considering the low profit margins in the fresh produce industry [18], the application of a carbon tax would strongly impact VFS’s economic viability by increasing the price of VFS product by as much as 23%. Hence, it is critical and urgent for the industry to address the energy and carbon challenges to ensure its economic and social development.

Another valuable output of LCAs is the identification of “hot spots” in a system, meaning that the parts of the lifecycle that have the largest environmental impact (Figure 1). A meta-analysis of 47 urban agriculture LCA reports found that (at least for indoor hydroponic farms) on-farm operations such as energy consumption for lighting, temperature control, and irrigation were the highest CO_2_-emitting processes [17]. This finding is consistent with a recent UK government report on the carbon assessment of buildings, where the operational stage of an average building’s life cycle constitutes 80 to 90% of carbon emissions [19]. When breaking down these operational expenses by activity types, the latest industry survey found that lighting represented the highest energy use (55%), followed by cooling/vents (30%) and heating (11%; [13]). Taken together, these data prompted us to review approaches that address both energy sourcing and operational light consumption as two important approaches to reduce GHG emissions and limit the use of electricity over the lifetime of the VFS.

### 1.2. Electrification of VFS by Renewable Energy Sources Provides an Opportunity to Contribute Sustainably to Food Production

Although VFS still represents a very small proportion of the agricultural food footprint and output (approx. 0.02% of the UK’s 17 million hectares of farmland [20]), rapid technological innovation coupled with strong socioeconomic drivers are increasingly promoting this farming method [9]. The energy-intensive nature of VFS has contributed to a misperception that this farming is necessarily detrimental to the environment. However, because most of this energy is in the form of electricity, VFS stand to benefit substantially from the renewables revolution. For instance, in the UK, wind energy represents an increasing proportion of electricity production, reaching over 20% of the country’s annual mix in 2021, with daily fluctuations between 10% and 60% [21]. By using renewable energy as the main source of energy supply, VFS becomes the cleanest way of farming in comparison with traditional or greenhouse farms using a more conventional energy mix. A 2018 analysis by OneFarm illustrated an extreme example whereby, if all the energy of indoor farms came from carbon-free renewable sources, this operation would be four times less impactful on the climate than open field agriculture [22].

Beyond ensuring a low-carbon electricity supply and maximising the efficiency of consumption, VFS also have the potential to present themselves as a “shiftable load” [23] to the electricity grid. This type of demand side management can be deployed in complex, time-critical industrial processes such as those in the steel industry, where power consumption schedules can be varied, provided that the final product falls within acceptable tolerances [24,25,26,27]. It is feasible that VFS can also utilise such process windows to ensure that a desirable crop phenotype can be produced at an economically viable growth rate. Given the dominance of lighting in the energy footprint of indoor growing [17], an enhanced temporal understanding of photobiology will help to define how demand flexibility can be used within VFS to offer an additional resource to the grid. Such an understanding could also elucidate the acceptable trade-off in quality that can be made in return for the financial incentive of cheaper energy during periods of surplus [24]. The positioning of VFS as flexible assets in a smart electricity grid will enhance the viability of the industry and offer societal benefits beyond those conferred by their produce alone. With this in mind, we now review biological strategies which might help to define the process windows that allow us to realise this ambition.

## 2. Photobiology Strategies for Maximising Energy Efficiency and Yield

One of the key benefits of VFS is that environmental conditions such as light, temperature, and water use can be precisely regulated to promote crop growth. While significant advances have been made in refining the engineering processes utilised in VFS [28], optimising biological strategies for VFS has proven substantially more complicated. Since plant development can be engineered by controlling the environmental conditions, growers will likely need to consider the desired traits of their crop before defining the environmental traits of their VFS. Variables including lighting and temperature, as well as atmospheric parameters including humidity, ventilation, and CO_2_ concentration, are crucial to consider in VFS and will certainly contribute to optimising yield. The complex interactions between lighting and these other variables may only be elucidated by first acquiring a deeper understanding of each contributing factor. In this review, we aimed to consolidate a small portion of this complex parameter space by exploring how our understanding of plant photobiology can be utilised to optimise growth within VFS (Figure 2).

### 2.1. Daily and Seasonal Variations in Light Irradiation Are an Intrinsic Component of Plant Biology

The rotation of the Earth and its axial tilt induce regular environmental changes including day/night cycles and seasonality. The predictable nature of these changes ensures that there is a competitive benefit in anticipating these environmental transitions, especially in photosynthetic organisms such as plants [29]. These benefits have driven the evolution of the circadian system, which enables environmental signals to be modulated in reference to dusk and dawn. As a consequence, it is necessary to consider time as an intrinsic part of the designed environment in VFS. Some of these aspects are obvious: seedlings are more likely to establish and grow during the spring than in the depths of winter, and daylength and temperature can be set accordingly. While it is unlikely that agronomists would routinely choose to replicate winter during cultivation, there may be specific environmental signals that promote the desired traits of a crop. The circadian system regulates flowering, photosynthesis, and stress responses, which are significant factors for crop yield and quality [30,31]. It is therefore informative to consider how plants integrate light signals (along with photosynthesis) over daily timescales to maximise growth, since each of these variables provides opportunities to improve crop performance and, consequently, limit energy use.

In horticultural lighting, three dimensions should be considered: light intensity, the light spectrum, and photoperiod. Each of these components influences plant photomorphogenesis and photosynthesis, and it is readily apparent that bespoke lighting regimes will alter the traits of the desired crop (e.g., leaf, seed, or secondary metabolites [30,32]). Light intensity, or the photosynthetic photon flux density (PPFD), defines the number of photosynthetically useable photons emitted in a certain area in a certain period of time, which is always expressed in μmol m^−2^ s^−1^. Light quality refers to the specific combination of different wavelengths of light emitted by the light source. Meanwhile, photoperiod is the duration for which plants are exposed to light in a 24-h period. The daily light integral (DLI) can be obtained by multiplying the photoperiod and PPFD to give the total sum of radiation during 24 h and can be used as a summary statistic, particularly as DLI is directly proportional to the amount of energy consumed per day [33]. A key consideration for energy efficiency is therefore to select a DLI that maximises crop yield, although it is also necessary to consider the efficiencies of energy conversion. LEDs’ energy efficiency varies dramatically in different portions of the spectrum, whereas photosynthesis is more efficient when using light from the red portion of the spectrum [34,35]. Additional complications are added by the ability of plants to respond to the prevailing environment during development. In the subsequent sections, we provide examples of how light regimes can be varied to promote growth, but caution that individual crops will likely require bespoke lighting regimes to maximise energy efficiency.

#### 2.1.1. Light Intensity

Since plants utilise light for photosynthesis, there is a direct correlation between the fluence rate and biomass accumulation at lower light intensities when photosynthesis is light-limited, although this relationship is lost as the fluence rates increase [36]. For example, under a 16 h d^−1^ photoperiod of red and blue mixed light, lettuce leaf dry weight increased in line with PPFD between 150 μmol m^−2^ s^−1^ and 250 μmol m^−2^ s^−1^. By contrast, when PPFD was increased to 300 μmol m^−2^ s^−1^, no additional increase in yield was observed [37]. Growers should therefore select a fluence rate that prevents the limitation of photosynthesis by light but does not provide excess light energy that cannot be productively used. The growth habit of the crop should also be considered, with the addition of inter canopy lighting enabling higher (or more uniform) fluence rates if necessary [30]. An additional consideration for the grower is whether to optimise their farm for maximal growth or energy efficiency, although this calculation will be greatly affected by the relative value of the crop and the economic and environmental cost of the available energy resources.

#### 2.1.2. Light Quality

Light quality also plays a vital role in plant growth. Although chlorophyll absorbs light across the visual spectrum (and so contributes a “broadband” light signal to the plant via photosynthesis), plants also utilise colour-specific light sensors to precisely adapt to the prevailing environment. These specific photoreceptors include phytochromes, cryptochromes, and phototropins, which enable plants to sense the different wavelengths of light [38,39,40]. Phytochromes are long-wavelength light receptors that regulate plant photomorphogenesis in response to red (600–700 nm) and far-red light (700–750 nm) [38,41]. Cryptochromes and phototropins are capable of detecting specific wavelengths of blue light (400–500 nm) [39,40]. Plants also perceive and respond to UV light, although the high energy of these wavelengths precludes their predominant use in VFS. Instead, UV light should be considered as a developmental cue rather than an energy source [42].

Fascinatingly, manipulation of these photoreceptors (using different colours of monochromatic LEDs) provides ample opportunity to manipulate crop growth beyond simply varying the number of photons provided as an energy source. Although plants will complete their lifecycle when grown under monochromatic light, such plants tend to be developmentally atypical because of the uneven activation of their photoreceptors [43]. Plants grown under monochromatic light consequently have dysfunctional photosynthesis and impaired biomass accumulation [44,45]. Cultivating crops under a combination of different wavelengths is therefore desirable. Economic factors will contribute to this decision, as fixed-wavelength lighting systems are typically less expensive than those with adjustable spectra. However, as the technology matures and additional colours of LED become commercially viable, there will be clear opportunities to design bespoke combinations of LED sources for use in VFS, depending on the crop and desired traits.

Lettuce is commonly grown in VFS and has consequently been used as an example to illustrate the effects of different qualities of light [45,46,47,48]. While there are variations between experiments, a greater proportion of red (R) relative to blue (B) light was beneficial, with increased yield and improved nutritional content reported. Supplementation of R + B mixed LEDs with green light (which can penetrate further into the canopy than R + B, and which contributes to the maintenance of circadian rhythms) can also promote lettuce leaf growth [49,50,51,52]. Similarly, the addition of far-red wavelengths to R + B conditions can also promote growth [53,54]. Such experiments demonstrate how additional complexity within the light regime can promote growth and nutrition quality, although, clearly, the economic trade-offs among these benefits, increased engineering complexity, and energy efficiency need to be considered.

#### 2.1.3. Photoperiod and the Control of Flowering

Beyond light intensity, changing the photoperiod provides an additional way to vary DLI. In a straightforward case, continuous light can be provided to maximise DLI, with correlated increases in yield. While this is achievable in some species (e.g., lettuce [55,56,57]), some crops (e.g., tomato) require a regular interval of darkness to prevent the onset of leaf chlorosis and necrosis [58]. Interestingly, the negative consequences of constant light upon tomato’s performance can be minimised by varying the light intensity or quality across a 24-h period [59]. These experiments illustrate how precise control of light regimes can promote the growth of individual species.

Despite the potential value of extending DLI by varying the photoperiod, the contribution of daylength to initiating the transition to flowering must also be considered. Inducing (or delaying) flowering significantly contributes to the yield and quality of crops, even though the coordination of flowering with seasonal variations is unnecessary in VFS. Lettuce is a long-day plant that flowers during the summer in the natural environment, whereas tomato is a day-neutral plant that flowers regardless of daylength. Previous reports demonstrated that basil and lettuce growth can be promoted by altering the daylength [28,60,61]. During vegetative development, meristems continue to propagate leaves but following the developmental transition to flowering, the meristem is transformed into a reproductive meristem that instead produces inflorescences and flowers [62]. Controlling the timing of the flowering transition can subsequently affect the yield and quality of plant leaves, stems, seeds, and fruits [63]. For leafy vegetable crops such as lettuce, the growth of plants in the vegetative development stage is more important for food production, with the floral transition preventing the initiation of new leaves. Conversely, the reproductive phase is more critical for fruit production in tomato. Flowering also initiates metabolic changes, with resources in the leaves being re-mobilised and transported to the developing reproductive tissues. The movement of sugars away from the leaves can result in a bitter taste in the leaves, negatively affecting crop quality [63]. The transition to flowering consequently has a significant impact upon crop production and should be considered when designing the optimum lighting regime for a particular crop.

### 2.2. Variable Lighting with Consideration of Biological Parameters Provides Opportunities to Coordinate Energy Loads with Renewable Energy Grids

In the natural environment, dawn and dusk mark predictable changes in irradiation, with these regular variations occurring in concert with cloud cover to affect incident light in an unpredictable manner. The circadian system assists with the interpretation of these signals in two ways: by enabling plants to anticipate and appropriately respond to daily and seasonal changes [64], as well as mitigating against irregular changes in irradiation [29]. Crops in VFS are typically grown in consistent “square wave” light regimes, but whether light levels could be manipulated during the day to maximise efficient growth remains to be explored. Such manipulations could be designed from two perspectives, either optimising irradiation to maximise the crop’s photosynthetic performance throughout the day, or by optimising energy usage to take advantage of variations in energy costs and availability from the electrical grid. In either case, dynamic application of light will likely have commercial benefits for growers.

Life cycle assessment reveals how lighting and temperature control are the major contributors to energy consumption throughout the operation of a VFS, where the required energy can be up to six times the electricity used for lighting in greenhouses [65]. There are consequently strong commercial and environmental incentives to optimise energy use within VFS, with potential solutions spanning engineering and biological strategies. While technological advances continue to improve the efficiency of VFS’s power consumption, improvements within the biological sphere remain limited by the inherent variation of crops and the disparate traits desired by growers.

Although energy consumption is a significant factor in VFS operation, it is important to note that these energy demands need not be static. Our review demonstrates how varied light conditions across both daily and seasonal timescales can influence crop growth. Exploiting this knowledge in electrically illuminated environments may facilitate intelligent, dynamic utilisation of excess energy available on the renewable energy grid and curtail consumption without detriment to crops when renewable sources are less forthcoming. It is crucial, however, that these variations in energy consumption are made with consideration of the crop being grown. Data gathered across numerous studies demonstrate how crop yield and quality can be promoted by considered selection of the lighting conditions [30]. It is equally apparent that the individual requirements of each crop (and the specific commercial traits desired) will drive the selection of a preferred light regime. Selection of daylength has particular consequences upon flowering time, and acceleration or delay of flowering can be precisely controlled. Extending the flowering time of leafy green vegetables allows these crops to have more time to grow leaves, helping to increase yield and promote nutrient accumulation. By contrast, accelerating flowering in fruit crops allows shorter generation times, reducing the cost of cultivating fruits. Although less well understood, it is also possible to use different qualities of light to modulate crops’ secondary metabolism, thereby altering the taste and quality characteristics [30]. Since light quality also affects plant architecture, it is also likely that crop uniformity can be promoted simply through the rational utilisation of light.

### 2.3. Selective Breeding Provides Opportunities to Optimise Crops for VFS

Plant domestication has occurred multiple times throughout history, and frequently incorporates selective breeding to promote common desirable traits such as increased fruit/seed size and a reduction in bitter-tasting metabolites [66]. Beyond these improvements for edibility, breeding often aids farming by inducing a loss of seed dormancy, synchronising flowering time, and reducing photoperiodic sensitivity. This latter trait is of particular importance as crops are moved away from their ancestral growing regions and are subsequently grown at different latitudes [67,68].

In the natural environment, light and temperature, in particular, vary in concert throughout the year. These regular environmental signals have driven the evolution of complex sensory pathways that enable plants to respond to daily and seasonal changes. Natural variation within a population is often beneficial, as it enables communities to accommodate weather patterns and climate change. Since plants interpret the photoperiod by comparing the daylength to signals born of the circadian system, it is perhaps understandable that the genes responsible for circadian function have been selected during the domestication of soybean, tomato, sugar beet, rice, and barley [31,68]. Such commonalities demonstrate the conservation of the circadian system and highlight future avenues of research in the pursuit of crops optimised for VFS production, where photoperiod can be adjusted at will.

The fully controlled indoor environment provided by VFS enables the planting and growth of crops independently of natural constraints. While many commercially relevant varieties can simply be moved from the field into VFS, it is likely that these contain genetic traits with unnecessary variation. Previous domestication and selective breeding efforts have proceeded through numerous genetic bottlenecks that did not consider growth under controlled lighting as a selection parameter, and there is therefore an excellent opportunity to specifically select crops which will thrive in VFS. Desirable traits will extend beyond those related to commercial value to facilitate automated harvesting processes, as well as making rational choices to introduce genetic alleles that complement the available light regimes. Future breeding efforts to produce crops optimised for VFS will also need to standardise crops’ responses to environmental signals [30,66]. Beyond these commercial requirements, it remains difficult to predict how individual varieties will respond to growth in VFS. Our understanding of plant photobiology provides tools to specifically target likely causes of this variation to produce a consistent crop. In addition, domestication inevitably restricts genetic diversity through propagation of individual plants and therefore “bioprospecting” within a crop’s genetic lineage will likely complement targeted approaches to improve crops’ suitability for VFS.

## 3. Conclusions

In this review, we have examined how an understanding of plant biology might accommodate the integration of renewable and fluctuating energy sources (such as wind) into VFS by varying crop irradiation. The dynamic regulation of lighting has two benefits: it allows crop growth in environments closer to their ecological (or domesticated) niche whilst enabling the coordination of energy usage with availability within the electrical grid. Utilisation of VFS in this manner will provide an additional component of the evolving smart grid as we seek to integrate renewable energy sources [23,24,69]. The adoption of renewable energy sources will dramatically reduce the global heating footprint of VFS, enabling the expansion of this developing sector.

The controlled environment of VFS also facilitates the adoption of additional technologies to promote crop growth. Remote assessment of crops’ photosynthetic performance will allow lighting regimes to be adjusted to maximise efficiencies, while machine learning will enable the coordination of light, temperature, and humidity parameters [70]. In this way, we expect VFS to serve as a proving ground for technological advancements as we seek to integrate new capabilities into the agricultural and horticultural industries.

## Figures and Tables

**Figure 1 biology-11-00922-f001:**
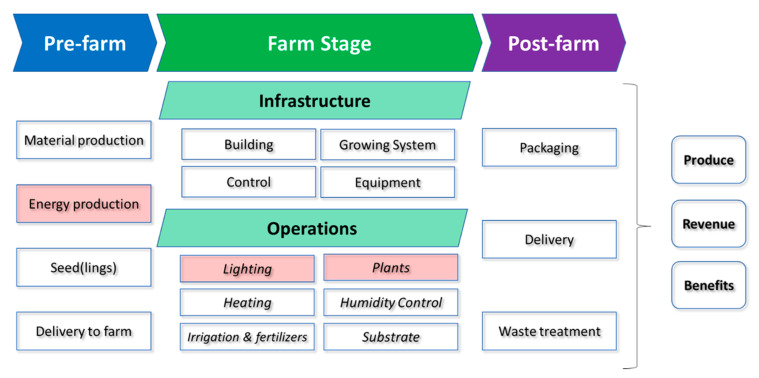
Vertical farming system activities organised in a life cycle assessment process diagram to highlight the carbon costs in controlled environment agriculture. Groups highlighted in red represent activities with high environmental impact that are addressed in this review.

**Figure 2 biology-11-00922-f002:**
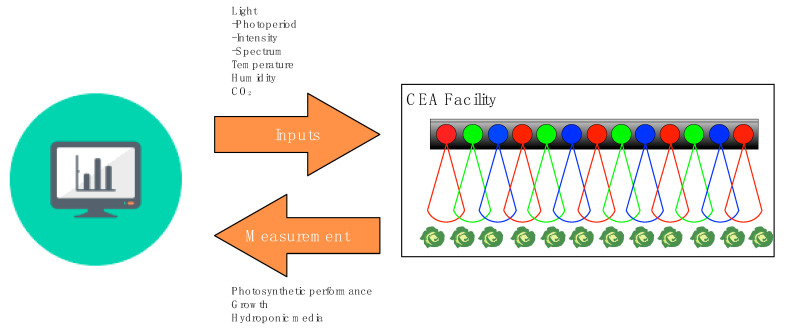
Controlled environment agriculture enables precise control and monitoring of the growth environments. Light properties that can be varied include photoperiod (daylength), intensity, and quality. These factors interact with other variables (including temperature, humidity, and CO_2_ concentration), which are not considered in this review. Future progress in this field will be accelerated by the improved measurement of photosynthetic performance, crop quality indicators, and nutritional media composition, which could be combined with machine learning to further improve lighting regimes.

## Data Availability

Not applicable.

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
