# Peer review of "Realising the Environmental Potential of Vertical Farming Systems through Advances in Plant Photobiology"

_biology, 2022, doi:10.3390/biology11060922_

Round 1

Reviewer 1 Report

Abstract: Replace ‘One solution to mitigate…’ with ‘One potential solution to mitigate…’.

Keywords: Photobiology is already in the title.

Perhaps this is a cultural difference, but in the US we typically use the term ‘global warming’ (not global heating).

I question the validity of the statement ‘most GHG emissions originate after departure from the farm-gate’. What about the production and use of synthetic fertilizers and their environmental impacts?

Do you have a more recent reference for the global population by 2050?

Why don’t you mention that VFS often are located in/near cities on properties with very high land prices?

Figure 1: Seedlings (of leafy greens) are often produced on the farm. Farm State, Infrastructure: Use building, growing system, control and equipment (four boxes). Operations: Add humidity control. Should labor be included in this diagram?

The carbon impact of a crop can also be normalized by dividing it by yield per unit volume or yield per unit production area.

Confusing: An average of 38.8 kWh and 5.4 kWh per kg or product.

You combine greenhouse production with traditional farming, but greenhouses can also be operated on renewable energy (making them also a ‘clean’ production system).

Your statement ‘plant development is inherently plastic’ is neither helpful nor convincing without additional explanation.

Your statement ‘are not trivial but are omitted here for the sake of brevity’ conveniently exclude a very important aspect of photobiology. This omission severely reduces the scientific value of your manuscript.

Figure 2: Explain that with light quality you refer to the spectrum? What do the arrows represent? Control and monitoring?

Where are the references for your statement that ‘it is readily apparent that bespoke lighting regimes…’?

Light intensity: Why do you not also discuss light uniformity (horizontally and vertically; the latter especially for taller crop canopies)?

Light quality: Why do you not include the most recent discussion about ePAR (extended PAR; 400-750 nm)? Far-red light is not defined as 700 nm and above.

I question the statement ‘commercially available lighting systems are typically available at discounted rates’. This is certainly not true in the US.

What spectrum do you recommend for lettuce production (you talk about the spectrum, but do not give a recommendation).

As you point out, most VFS produce leafy greens. However, no leafy greens are ever produced under conditions that induce flowering in VFS (it would make the crop unsalable). Therefore, the value of discussing photoperiod and flowering is questionable.

The impact of sun flecks on plant production is minimal and probably not of immediate concern for the topic discussed in your manuscript.

What do you not discuss the potential benefits of using light fixtures with higher efficacies? Similarly, energy savings might be feasible by using pulse width modulation of the electric light delivered.

A statement like ‘The required energy for VFS is approximately six times the electricity used in greenhouses’ is useless since the factor depends on the crop and the site (amount of solar radiation).

How does your review ‘demonstrate how varied light conditions… can benefit crop growth’?

Again, the ‘flowering time of leafy green vegetables’ is not something growers are concerned about.

Domestication is not the same as breeding. I think you mean breeding instead of domestication.

The staple crops you mention are not typically produced in VFS.

The term artificial lighting is incorrect. Suggest to use electric lighting.

What are ‘land races’?

Is the first sentence of your conclusions a true statement?

Please format your references consistently (e.g., some paper titles are all caps).

Reviewer 2 Report

This is a review paper with an emphasis on photobiology. But the authors also extended to other fields, for example, renewable energy, breeding, etc. I would like to see if the authors could elaborate a little bit about renewable energy to be used for VFS to reduce energy costs. As it is well known and indicated by the authors as well, the most expensive operational costs are electricity (and labor). The term "renewable energy" appears two times in subtitles and in the concluding remarks. However, there is little information in the text about what/how renewable energy can be used for VFS.  what are the specific renewable energy sources that are most likely to be used for VFS? what are the challenges and hurdles at this time to apply renewable energy in VFS? 

In Figure 2, what do the authors try to demonstrate? 

Reviewer 3 Report

line 57. Water use depends on the cultivation technology, eg NFT gives similar results in a greenhouse as VFS. So it is not the place that matters, but the cultivation technology.

line 92. For the sake of completeness, it would be good to report the amount of CO2 assimilated by plants per kg of product, not just the CO2 emissions when producing electricity.

Line 207 (chapter 2.1.2) When talking about the quality of light, it is necessary to discuss the McCree curve (doi 10.1016 / 0002-1571 (71) 90022-7), which describes the influence of the light wavelength on a quantum yield of photosynthesis. In addition, the efficacy of LEDs must be considered, which varies greatly depending on the light quality and LED manufacturing technology and has so far mostly been in the range of 2-3 umol/J (doi:10.3390/su12187516). Recent reports state that the efficacy of red LEDs exceeds 4 umol/J (https://www.osram.com/os/applications/horticulture-lighting/index.jsp). This has a large impact on energy use per kg of product, since, for example, 2 kWh of electric power is needed at 2 umol/J led efficacy, 16 h and a PPFD of 250 umol/m2/s (DLI 14.4 mol/m2) and only 1 kWh at 4 umol/J efficacy. Expanding the color of the leds may improve yield but significantly increase electricity inputs. For example, reports that green light increased plant weight usually do not take into account electric power consumption. In the case of green, it is very energy inefficient due to the low efficacy of green led and the low quantum yield of photosynthesis (McCree curve). Nevertheless, the information that green light can improve growth may be the beginning of research to increase the efficacy of green leds.
